# Gut microbiome diversity is associated with sleep physiology in humans

**Robert P. Smith[1], Cole Easson[1,2], Sarah M. Lyle[3], Ritishka Kapoor[3], Chase P. Donnelly[1], Eileen J. Davidson[1], Esha Parikh[3], Jose V. Lopez[1], Jaime L. Tartar [3]***

**1** Department of Biological Sciences, Halmos College of Natural Sciences and Oceanography, Nova Southeastern University, Fort Lauderdale FL, United States of America, **2** Biology Department, Middle Tennessee State University, Murfreesboro, TN, United States of America, **3** Department of Psychology and Neuroscience, Nova Southeastern University, Fort Lauderdale, Florida, United States of America

* tartar@nova.edu

**Data Availability Statement:** All raw data for the findings in this article can be found at: 10.6084/m9. figshare.9765227.

## Abstract

The human gut microbiome can influence health through the brain-gut-microbiome axis. Growing evidence suggests that the gut microbiome can influence sleep quality. Previous studies that have examined sleep deprivation and the human gut microbiome have yielded conflicting results. A recent study found that sleep deprivation leads to changes in gut microbiome composition while a different study found that sleep deprivation does not lead to changes in gut microbiome. Accordingly, the relationship between sleep physiology and the gut microbiome remains unclear. To address this uncertainty, we used actigraphy to quantify sleep measures coupled with gut microbiome sampling to determine how the gut microbiome correlates with various measures of sleep physiology. We measured immune system biomarkers and carried out a neurobehavioral assessment as these variables might modify the relationship between sleep and gut microbiome composition. We found that total microbiome diversity was positively correlated with increased sleep efficiency and total sleep time, and was negatively correlated with wake after sleep onset. We found positive correlations between total microbiome diversity and interleukin-6, a cytokine previously noted for its effects on sleep. Analysis of microbiome composition revealed that within phyla richness of Bacteroidetes and Firmicutes were positively correlated with sleep efficiency, interleukin-6 concentrations and abstract thinking. Finally, we found that several taxa (*Lachnospiraceae*, *Corynebacterium*, and *Blautia*) were negatively correlated with sleep measures. Our findings initiate linkages between gut microbiome composition, sleep physiology, the immune system and cognition. They may lead to mechanisms to improve sleep through the manipulation of the gut microbiome.

## Introduction

The human gut microbiome can exert effects on mental and physical health through different routes including through the brain-gut-microbiome axis (BGMA [1]), intestinal activity [2], and the competitive exclusion of pathogenic bacteria [3]. BGMA signaling in particular has

**Funding:** This research was supported by a Presidents Faculty Research and Development Grant #335411 through Nova Southeastern University awarded to JLT. This program played no role in the study design, data collection and analysis, decision to publish, or preparation of the manuscript.

**Competing interests:** The authors have declared that no competing interests exist.

been shown to be bi-directional, where not only can gut bacteria influence health and behavior, but psychological states can alter gut health. Perturbations to the BGMA have been associated with gastrointestinal disorders [4], depression and mental quality of life [5], Parkinson's disease [6], increased anxiety [7], and decreased cognitive abilities [8]. While the mechanisms through which the gut microbiome and human body interface have yet to fully understood, previous work has shown that bacteria can influence neural [9], hormonal [10] and immune responses [11], and permeability of both the gut [12] and the blood brain barrier [13]. Accordingly, understanding how the BGMA functions to regulate human health and behavior is of importance.

Several bacterial metabolites have been identified as possible mechanisms through which bacteria communicate via the BGMA with their host. Chief amongst these are metabolites that interface with the immune system [14]. For example, short chain fatty acids (SCFA, e.g., butyrate, acetate) produced by fermenting bacteria can suppress pro-inflammatory cytokines, and interact with regulatory T cells to attenuate colitis [15]. The bacterial metabolite indole stimulates the production of interleukin-22 (IL-22), which stimulates the production of anti-microbial peptides thus serving a protective role against pathogens [16]. Polysaccharide A downregulates the production of the pro-inflammatory IL-17, while upregulating the production of IL-10, which together serve to protect against colitis [17]. The production of IL-6 and IL-1β can be stimulated by the gut microbiome, which can lead to regulatory B-cell differentiation [18]. Overall, there are well-established links between the immune system and the gut microbiome in humans.

Sleep is a physiological state that is intrinsically linked to the immune system but is overall understudied in the context of BGMA. In general, short sleep duration and poor sleep quality have been associated with several aspects of cognitive and neurobehavioral performance [19–21], and several diseases including cancer [22], type II diabetes [23], and Alzheimer's disease [24]. Notably, cytokines represent a potential critical interface between sleep physiology and gut microbiome composition. The acute phase pathway cytokines IL-1β and IL-6 in particular are strongly associated with sleep physiology. IL-1β is a major somnogenic factor [25–27]. IL-1β administration in human and non-human animals increases spontaneous sleep and fatigue, and IL-1β increases with ongoing sleep loss [27, 28]. Unlike IL-1β, IL-6 is not a direct somnogenic factor, but sleep loss results in increased IL-6 levels [29]. In the gut, IL-6 and IL-1β mediated-inflammation fluctuate in response to stress and disease [30, 31]. For example intestinal mucositis results in increased expression of IL-6 and-IL-1β in the small intestine [32, 33] and in serum and colon tissue [34] in mice. In humans, chronic stress alone increases IL-6 and-IL-1β [35].

Despite the close relationship between cytokine activity, gut microbiome activity and sleep, only a handful of studies have examined sleep and gut-microbiome composition. In mice, periods of intermittent hypoxia, which serves to simulate obstructive sleep apnea [36], and sleep fragmentation, have been shown to alter the gut microbiome diversity [37]. In humans, previous research has shown that partial sleep deprivation can alter the gut microbiome composition in as little as 48 hours [38], however longer periods of sleep deprivation apparently do not have this effect [39]. A more recent study showed that high sleep quality was associated with a gut microbiome containing a high proportion of bacteria from the *Verrucomicrobia* and *Lentisphaerae* phyla, and that this was associated with improved performance on cognitive tasks [40]. In spite of these findings, the mechanisms through which the gut microbiome can affect sleep remains unresolved, and in particular, the molecules that interface between sleep and the gut microbiome remain unidentified. To address this uncertainty, we investigated the relationship between gut microbiome diversity, sleep, cognition and the pro-inflammatory cytokines, IL-6 and IL-1β. To accomplish this, we used a multidisciplinary approach consisting of

microbiome sequence, actigraphy, cognitive and neurobehavioral testing, and biochemical approaches to measuring immune system markers.

## Materials and methods

### Participants

Recruitment and testing procedures were approved by the Nova Southeastern University (NSU) Institutional Review Board (IRB). All participants received a verbal explanation of the study procedures and signed an NSU IRB-approved written Informed Consent Form. Forty male participants were recruited. Participant recruitment and testing occurred between May of 2017 and March of 2018. One participant was excluded from our analysis due to non-compliance during testing. Individuals self-identified as having taken pharmaceuticals or with a past history of gastrointestinal illness were excluded from analysis. We excluded these individuals as previous work has shown that pharmaceuticals (e.g., [41] and gastrointestinal illnesses (e.g., [42]) can drastically alter gut microbiome composition. As such, a total of 26 participants (n = 26, 26 males, μ = 22.19, standard deviation = 3.11) were used for final analysis. Two participants were not compliant with Actiwatch (Philips Medical Systems, Miramar, FL) procedures, and therefore sleep data was not collected from these participants. Two participants failed to provide a sufficient fecal sample for genomic sequence, and thus microbiome data was not collected for these participants. Participants were compensated using a $50 gift card.

### Procedures

To control for circadian variation in cortisol and immune system markers, testing procedures occurred between 2–4 pm. Following consent, the height and weight of participants were measured (average Body Mass Index (BMI) = 25.0, SD = 3.3). Participants then completed the NIH Toolbox (neural-behavioral measurements, Bethesda, MD) and the Joggle Research platform (cognitive testing, Seattle, WA) using a supplied iPad (Apple, Cupertino, CA). 1 mL of saliva was collected into a 1.5 mL polyethylene centrifuge tube using a passive drool technique using a small sterile cylinder in order to measure selected biomarkers, outlined below. Saliva was immediately stored at -20˚C. Finally, to characterize the gut microbiome, each participant was provided a sterile fecal swab (Health Link, Jacksonville FL) to collect fecal matter. Self-collection of fecal matter occurred within 12 hours of neurobehavioral testing. Upon collection, fecal swabs were immediately stored at -20˚C.

### Actigraphy

Participants were required to wear an Actiwatch for 30 days after testing, upon which the Actiwatch was returned and the data were recorded and analyzed. Measurements included bed time (average), get up time (average), time in bed (hrs), total sleep time (hrs), onset latency (mins), sleep efficiency, wake after sleep onset (WASO, mins), and number of awakenings.

### Neurobehavioral testing

Neurobehavioral testing was conducted using the automated "Cognition" test battery from Joggle Research (Joggle Research, Seattle WA) and the Emotion test battery from the NIH Toolbox (Health Measures, Northwestern University, IL). The Joggle Cognition battery consists of eight cognitive measures administered on a standard electronic tablet (Apple IPad). Total testing time is approximately 20 minutes, which prevents participant fatigue. The cognition test battery consists of eight tasks covering a diverse set of cognitive domains (e.g. executive function, episodic memory, complex cognition, and sensorimotor speed) and are based on

tests known to activate specific brain systems[43]. The tests include a Psychomotor Vigilance Test (PVT), the Balloon Analog Risk Task (BART). the Digital Symbol Substitution Task (DSST), the Line Orientation Task (LOT), an Abstract Matching (AM) test, the NBACK, a Visual Object Learning Task (VOLT), a Motor Praxis Task (MPT). The NIH Toolbox Emotion measures include four major domains: Psychological Well-Being, Stress and Self-Efficacy, Social Relationships and Negative Affect. Specific subtests include measures of: Anger, Fear, Depressive Symptoms, Psychological Well-Being, Positive Affect, General Life Satisfaction, Meaning & Purpose, Perceived Stress, Self-Efficacy, Social Support, Emotional Support, Loneliness, Friendship, and Social Distress [44].

## IL-1β, IL-6 and cortisol

Saliva samples were run in duplicate and quantified via a human enzyme immunoassay (ELISA) kit as per the manufacturer's instructions (Salimetrics LLC, USA). Upon, thawing, samples were vortexed and centrifuged for 15 min at 1,000 x g. The samples were immediately read in a BioTek ELx800 plate reader (BioTek Instruments, Inc., USA) at 450 nm with a correction at 630 nm. All samples were within the detection ranges indicated in the immunoassay kits, and the variations of sample readings were within the expected limits. Final concentrations for the biomarkers were generated by interpolation from the standard curve in μg/dL for cortisol (sensitivity = <0.007, range 0.012–3.000 ug/dL) and pg/mL for IL-1β (sensitivity = <0.37 pg/mL, range 3.13–200 pg/mL) and IL-6 (sensitivity = 0.07 pg/mL, 0–100 pg/mL).

## Next generation sequencing and analysis

Total genomic DNA was extracted from one of the preserved replicate swabs using the MoBio BioStic kit following the manufacturer's protocol. After extraction, polymerase chain reaction (PCR) was used to amplify the V4 region of the 16S rRNA gene using the primers and protocols established by the Earth Microbiome Project [45, 46]. During PCR, each sample was given a unique twelve base pair Golay barcode. PCR products were cleaned with AMPure beads adhering to the Illumina protocol (REF), and the cleaned amplicons were checked on a Tapestation bioanalyzer to verify amplicon size. Amplicon concentrations were assessed using a Qubit fluorometer (ThermoFisher Scientific, Waltham, MA), and all samples were normalized to 4 nM before loading. Sample preparation and loading followed standard Illumina protocols for amplicon sequencing. The normalized amplicons were sequenced on an Illumina MiSeq (Illumina, San Diego, CA) sequencer using a 500 cycle V2 chemistry kit, which produced paired-end 250 base pair sequences.

Sequence processing was done in QIIME [45] and R [47]. Initially, forward and reverse sequences were separated into individual files using QIIME. The DADA2 pipeline was used for bioinformatics processing in R [48]. Sequences were trimmed to remove ambiguous bases (max N = 0), amplicons longer than 250 base pairs, and amplicons shorter than 160 base pairs. The default parametric error model in DADA2 was used to calculate sequence error rates. Next, sequences were dereplicated to infer sequence variants, forward and reverse ends were merged, chimeras were removed, and the sequence table was composed. The taxonomy of each sequence variant was determined using the Silva database (Release 128, [49]).

## Statistical analysis

The SPSS statistical package (version 19, SPSS Inc., IBM Company, Armonk, NY) was used to determine Pearson correlation coefficients (2-tailed) between physiological, neuro-behavioral, cognitive and microbiome diversity. Pearson correlation coefficients (with $P \leq 0.05$) were used to create a network diagram in Cytoscape (version 3.7, [50]). All raw data for the findings

in this article can be found at: 10.6084/m9.figshare.9765227. To create the network, we collected variables significantly correlated with microbiome diversity. This first set of correlated variables (predominantly sleep efficiency, abstract matching, and IL-6) were then used to identify additional correlated variables not directly correlated with microbial diversity. Once these variables were identified, we then examined correlations within all nodes in the network. Each Pearson correlation coefficient was added as a weight to each edge in the network, the value of which was indicated using the color of the edge (darker red = -1, darker blue = +1). Statistical analysis of microbiome data was conducted using the vegan package in R [51]. Prior to analysis, sequence abundance was transformed to relative abundance. Correspondence between microbiome composition and psychological metrics was assessed using a redundancy analysis and the goodness of fit for individual bacterial taxa was measured using an inertia decomposition analysis [51].

## Results

### Microbiome diversity is significantly and positively correlated with sleep efficiency

We found that all three measurements of microbiome diversity, richness ($\rho$ = 0.479, P = 0.001), Shannon diversity ($\rho$ = 0.643, P = 0.001), and inverse Simpson diversity ($\rho$ = 0.540, P = 0.009), were associated with sleep efficiency (Fig 1). While all three measures of microbiome diversity were negatively correlated with WASO, only Shannon diversity was significant ($\rho$ = -0.537, P = 0.01) as both richness ($\rho$ = -0.378, P = 0.083), and inverse Simpson diversity ($\rho$ = -0.395, P = 0.069) were not significant. All three measures of microbiome diversity were positively correlated with total sleep time. However, only inverse Simpson diversity was significant ($\rho$ = -0.443, P = 0.0039), whereas richness ($\rho$ = 0.284, P = 0.2) and Shannon diversity ($\rho$ = 0.380, P = 0.069) were not.

As a control analysis, we found that all three microbiome diversity measures were positively correlated with one another: (Shannon diversity with richness ($\rho$ = 0.873, P < 0.001), Shannon diversity with inverse Simpson ($\rho$ = 0.905, P < 0.001), richness with inverse Simpson ($\rho$ = 0.740, P < 0.001). Moreover, we found that sleep efficiency was positively correlated with time in bed (hrs, $\rho$ = 0.470, P = 0.020) and total sleep time (hrs, $\rho$ = 0.783, P < 0.001), and was negatively correlated with WASO ($\rho$ = -0.853, P < 0.001), and the number of awakenings ($\rho$ = 0.462, P = 0.023). WASO was positively correlated with the number of awakenings ($\rho$ = 0.462, P = 0.023).

### IL-6 is correlated with microbiome diversity and measurements of sleep

Given the ability of gut microbiome to interact with IL-1β and IL-6, we sought to understand if there were correlations between these two cytokines and measures of microbiome diversity, and sleep. We found that IL-6 was positively associated with microbiome richness ($\rho$ = 0.612, P = 0.001), Shannon diversity ($\rho$ = 0.508, P = 0.011) and inverse Simpson diversity ($\rho$ = 0.521, P = 0.009), thus demonstrating a link to microbiome diversity (Fig 1). Consistent with a previous report (48), IL-6 levels were positively correlated with time in bed (hrs, $\rho$ = 0.439, P = 0.032) and total sleep time ($\rho$ = 0.476, P = 0.019). While IL-6 had a positive correlation with sleep efficiency ($\rho$ = 0.344, P = 0.099), it was not significant. Similarly, there was a negative correlation with WASO ($\rho$ = -0.206, P = 0.334), but it was not significant. Finally, there was a slight but insignificant positive correlation between IL-6 and sleep fragmentation (measured via WASO) ($\rho$ = 0.042, P = 0.846). We observed no significant correlations between IL-

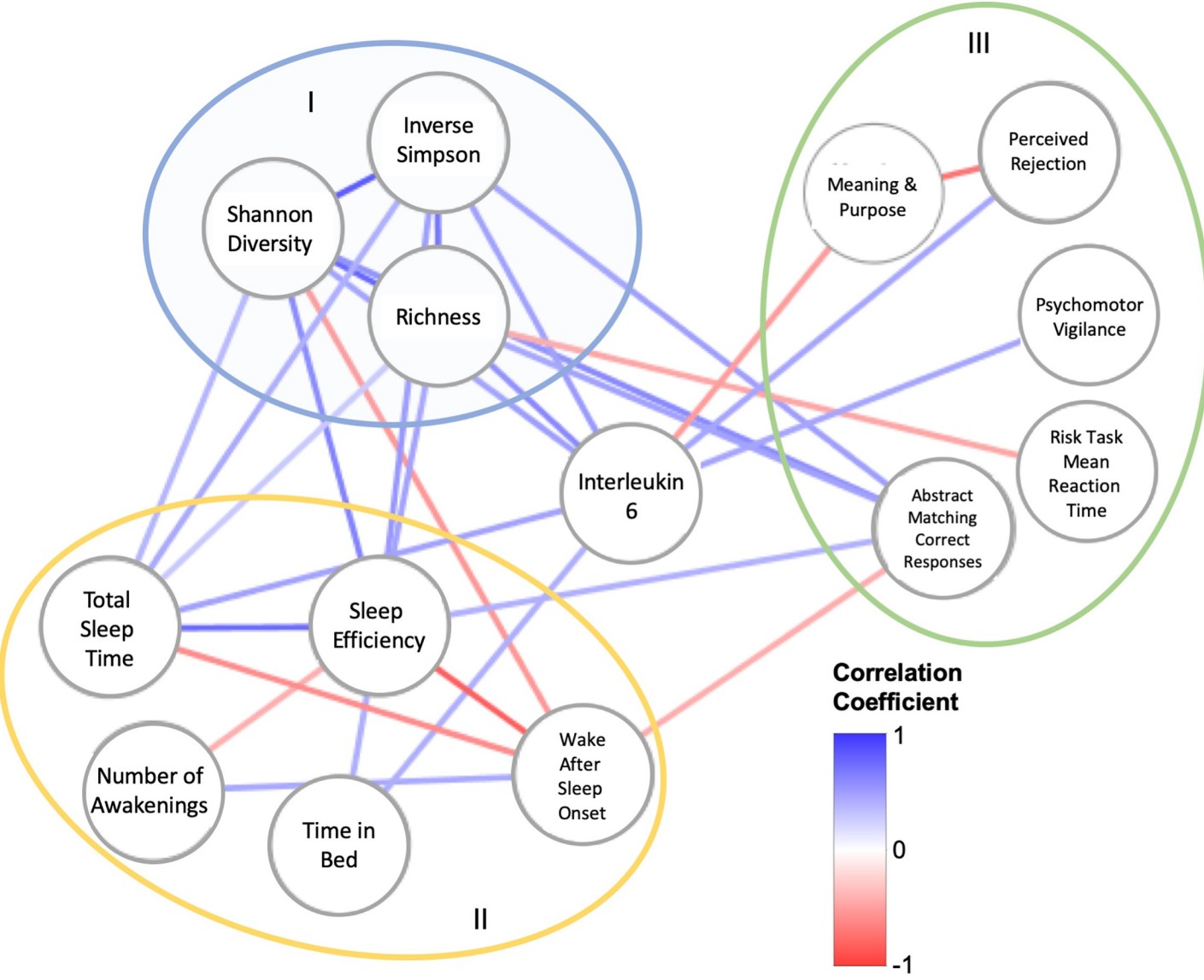

**Fig 1. The interaction network between measures of sleep, microbiome diversity and cognitive performance.** Pearson correlation coefficients were used to generate the weight of each edge in the network. Heat map shown on image. Different colored circles indicate groupings of nodes with similar traits in the network (I = microbiome diversity, II = sleep, III = cognition). Raw data for correlations (outside of microbiome diversity control correlations) found in S1–S4 Figs. Directionality of interactions is not implied in this figure.

1β and measures of sleep (S1 Table). Furthermore, we did not observe any significant correlations between cortisol, measures of sleep, and microbiome diversity.

## Specific metrics of cognition are impacted by microbiome diversity, sleep and IL-6

We found a large number of correlations between microbiome diversity, sleep and abstract matching (as measured using Joggle). Specifically, microbiome richness ($\rho = 0.489$, $P = 0.015$), Shannon diversity ($\rho = 0.607$, $P = 0.002$), and inverse Shannon diversity ($\rho = 0.501$, $P = 0.013$) were positively and significantly correlated with abstract matching (Fig 1). Moreover, sleep efficiency was significantly and positively correlated with correct abstract matching responses

(ρ = 0.443, P = 0.030), and was negatively correlated with WASO (ρ = -0.427, P = 0.037). There was not a significant correlation between IL-6 and correct abstract matching (ρ = 0.227, P = 0.265).

Beyond abstract matching, we found that psychomotor vigilance (measured using Joggle, ρ = 0.469, P = 0.016) and perceived rejection (measured using NIH toolbox, ρ = 0.451, P = 0.024) were significantly and positively correlated with IL-6. Working memory (measured using Joggle, ρ = -0.388, P = 0.045) and meaning and purpose (measured using NIH toolbox, ρ = -0.507, P = 0.010) were negatively correlated with IL-6. Perceived rejection and meaning and purpose were negatively correlated (ρ = -0.722, P < 0.001). Finally, we found that richness was significantly and negatively correlated with risk decision making (measured using Joggle, ρ = -0.461, P = 0.023).

## Bacteroidetes and Firmicutes in the gut microbiome are associated with sleep efficiency, IL-6, and abstract thought

We used redundancy analysis to determine any significant correlations between richness and diversity within bacterial phyla, and nodes in our interaction network (Fig 1). Significant and numerous correlations were observed in the Bacteriodetes phyla (Fig 2). Our analysis found a positive correlation between sleep efficiency and both the richness (ρ = 0.41, P = 0.05) and diversity (ρ = 0.45, P = 0.03) within the Bacteroidetes. Similar positive correlations were observed with IL-6 and abstract matching correct responses (richness and IL-6 (ρ = 0.66, P <0.001), diversity and IL-6 (ρ = 0.47, P = 0.02), richness and abstract matching (ρ = 0.45, P = 0.02), diversity and abstract matching (ρ = 0.56, P = 0.003)). Negative correlations between diversity and WASO (ρ = -0.49, P = 0.02), and richness and mean task reaction time (ρ = -0.39, P = 0.05) were also observed.

Similar trends were observed when examining the richness, but not diversity, within the Firmicutes phyla. Positive correlations between richness and sleep efficiency (ρ = 0.49, P = 0.02), IL-6 (ρ = 0.52, P = 0.01), and abstract matching correct responses (ρ = 0.51, P = 0.01) were observed. Similar to the Bacteroidetes, a negative correlation between richness and mean task reaction time (ρ = -0.5, P = 0.01) was also observed. Finally, we observed that richness within the Actinobacteria phylum was negatively correlated with the number of awakenings (ρ = -0.41, P = 0.05). Richness of the Proteobacteria phylum was positively correlated with IL-6 (ρ = 0.39, P = 0.05).

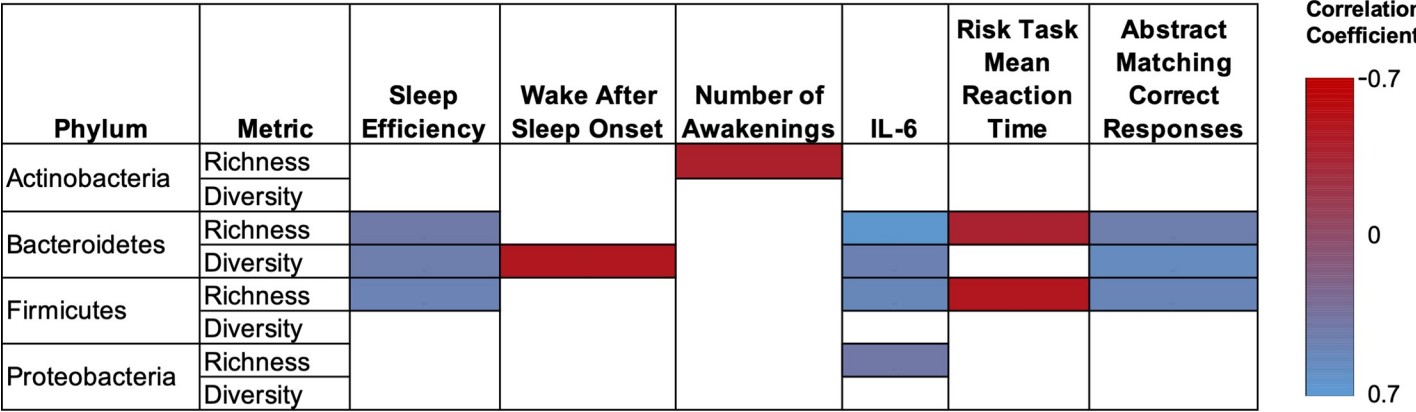

**Fig 2. The association of richness and diversity within bacterial phyla, and with measures of sleep, IL-6 and cognition identified in our interaction network.** Only Pearson correlations coefficients with P ≤ 0.05 are shown.

## Numerous taxa from the Firmicutes and Proteobacteria, but not Bacteroidetes, are associated with nodes in our interaction network

We used redundancy analysis to examine associations between bacterial taxa (identified primarily by genera) within the phyla identified above and nodes in our interaction network. Interestingly, despite broad level associations between Bacteroidetes and several nodes, our redundancy analysis failed to identify any significant associations between taxa within the Bacteroidetes and nodes in our interaction network. In contrast, we found significant correlations between 15 taxa in the Firmicutes and nodes in our network (Fig 3). For ease of reading, S2 Table contains all significant correlation coefficients and their respective P-values. Most notably, bacteria from *Blautia* sp., *Lachnospiraceae* (family), and *Oribacterium* sp., were generally negatively correlated with sleep efficiency and total sleep time. Exceptions to this included two different family members of the *Lachnospiraceae* that were found to be positively correlated with sleep efficiency and total sleep. Several taxa (*Geobacillus*, *Leuconostoc*, *Staphylococcus*, *Streptococcus*, *Tetragenococcus*) were positively associated with risk task mean reaction time. *Coprococcus* was positively associated with the number of awakenings. *Erysipelotricheaceae* and *Holdemania* were negatively associated with number of awakenings, and *Megamonas* was positively associated with risk task mean reaction time. Finally, members from the *Dialister* taxa were both positively and negatively associated with IL-6.

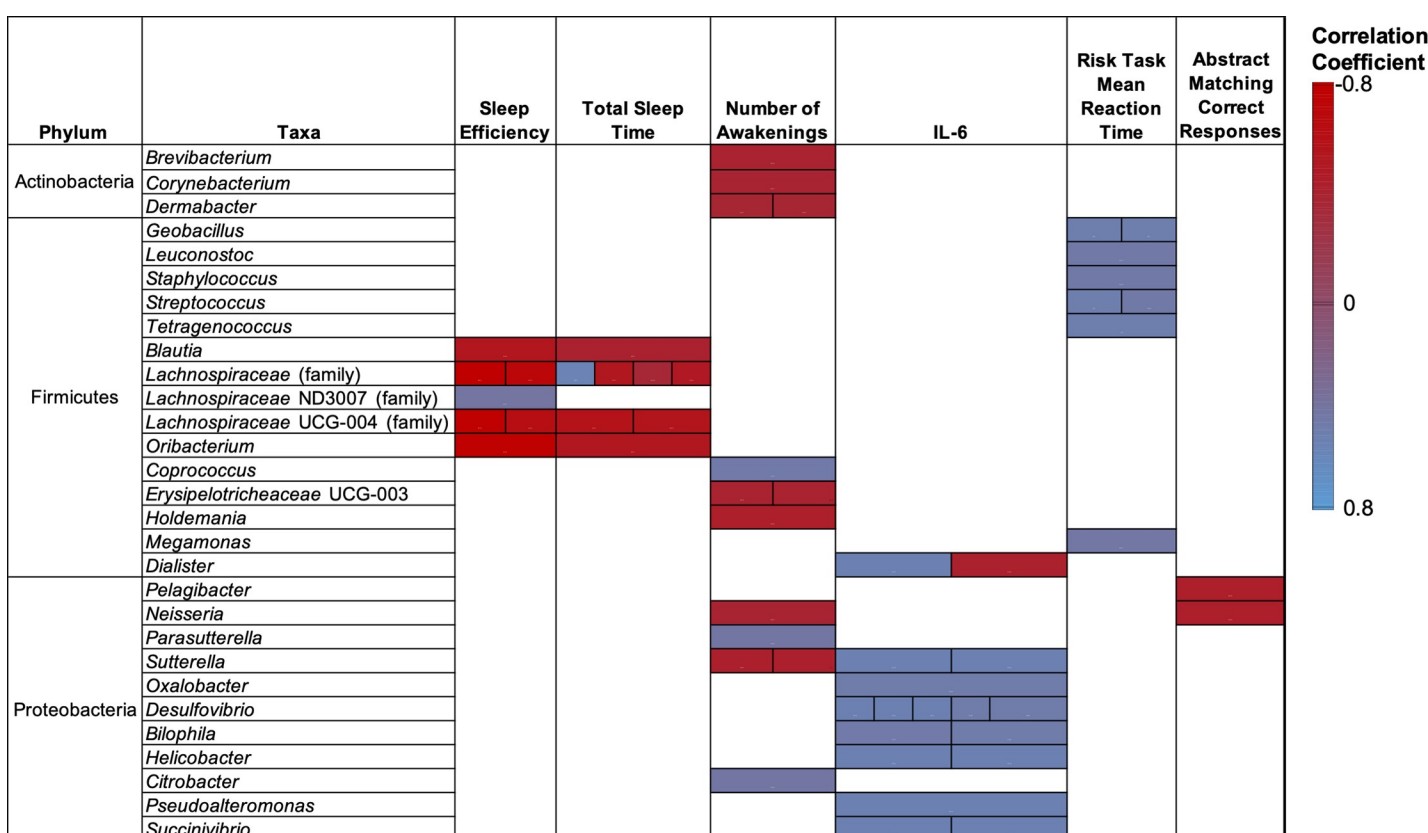

**Fig 3. Significant associations between bacterial taxa with measures of sleep, IL-6 and cognition identified in our interaction network.** Taxa were identified at the genus level unless otherwise indicated. Only Pearson correlations coefficients with P ≤ 0.05 are shown. Multiple boxes within the same column indicated significant associations between several operation taxonomic units (OTUs) and the node identified at the top of the column. Correlation coefficients and P values presented in S2 Table.

Three taxa from the Actinobacteria (*Brevibacterium*, *Corynebacterium*, and *Dermabacter*) were negatively correlated with the number of awakenings. Seven genera from the Proteobacteria (*Sutterella*, *Oxalobacter*, *Desulfovibrio*, *Bilophila*, *Heliobacter*, *Pseudoalteromonas*, and *Succinivibrio*) were positively associated with IL-6. *Neisseria* and *Sutterella* were negatively correlated with the number of awakenings. In contrast, *Parasutterella* and *Citrobacter* were positively associated with the number of awakenings. Both *Neisseria* and *Pelagibacter* were negatively associated with abstract matching correct responses. No additional significant correlations were observed between taxa with the phyla analyzed and nodes in our network. Taxa were identified at the genus level unless otherwise indicated. Only Pearson correlations coefficients with P < 0.05 are shown. Multiple boxes within the same column indicated significant associations between several operation taxonomic units (OTUs) and the node identified at the top of the column. Correlation coefficients and P values presented in S2 Table.

## Discussion

Disruption of sleep and sleep/wake functions have been associated with both short (e.g., increased stress responsivity, psychosocial issues) and long term (e.g., cardiovascular diseases, cancer) health consequences (reviewed in [52]). Despite well-established links between physical and mental health and sleep, disruption of sleep remains widespread. In 2017, 35% of Americans reported that their sleep quality was good, fair or poor [53]. Previous research has focused on understanding the psychological, societal, and physiological factors that regulate sleep. However, recent studies, including the current study, have found associations between sleep physiology and gut microbiome composition. To our knowledge, this is the first study to examine the associations between sleep, the immune system, and measures of cognition and emotion. A well-rounded understanding of how these facets of human physiology function may lead to a better understanding of the bidirectional communication between the host and the gut microbiome and may lead to novel sleep intervention strategies. For example, previous studies have demonstrated that fecal microbial transplants can improve disorders that are directly linked to the gastrointestinal tract (e.g., recurrent intestinal infections, ulcerative colitis [54, 55]). However, more recent work has demonstrated that such transplant strategies can alter aspects of human physiology that are not directly linked to the gastrointestinal tract but are instead conceivably linked via the BGMA. For example, fecal microbial transplants have been shown to improve cognition in patients suffering from cirrhosis [56], have altered behavior in individuals with autism [57], and have attenuated epileptic seizures [58]. While disruptions to sleep and sleep/wake functions are not considered to be gastrointestinal diseases, these recent studies indicate that fecal microbial transplants may represent a strategy to improve sleep efficiency via the BGMA.

We found that microbiome diversity (richness, Shannon diversity, and inverse Simpson diversity) was positively correlated with sleep efficiency, and total sleep time, and was negatively correlated with the sleep fragmentation (WASO). In other words, our results suggest that diversity of the gut microbiome promotes healthier sleep. This contrasts with two previous studies in humans that suggested that microbiome diversity is insignificantly affected following a period of sleep restriction [38, 39]. A critical difference between these studies and ours is that our study measured sleep over an extended period of time (one month) while the previous studies manipulated sleep by experimentally restricting sleep. Accordingly, it is possible that short-term manipulations to sleep do not influence the gut microbiome diversity, but rather that microbiome diversity can influence sleep in the long term.

We also found that IL-6 was positively correlated with the aforementioned measures of microbiome diversity, as well as total sleep time and time in bed. IL-6 is a putative somnogenic

factor in humans [59] and high daytime serum concentrations of IL-6 is associated with poor sleep quality [60, 61]. In addition, increased IL-6 levels are associated with fragmented sleep in mice [37]. IL-6 is also an important factor in sleep regulation as sleep onset often coincides with increased circulating IL-6 and IL-6 remains high during the night [62, 63]. While our study found clear associations between IL-6, gut microbiome diversity, and both bacterial phyla and taxa (Fig 2 and Fig 3), the mechanisms and/or metabolites that link these systems remains unknown. Interestingly, we failed to find a significant correlation between the general stress biomarker, cortisol, sleep measures, and microbiome diversity. As such, it appears the link between gut microbiome diversity and IL-6 is not mediated or influenced by stress despite the well-developed link between high IL-6 concentrations and stress [64]. Finally, we found that increased gut microbiome diversity correlates with abstract matching correct responses. The abstract matching test measures abstraction and flexibility components of executive function and reflects prefrontal cortex activity [43]. We note that a previous study that examined cognitive flexibility, sleep, and gut microbiome composition failed to find any significant influence between cognitive flexibility and gut microbiome composition [40].

Our results demonstrated that richness within the phyla Bacteroidetes and Firmicutes were positively correlated with sleep efficiency, while only the Bacteroidetes was negatively correlated with sleep fragmentation (WASO). These two phyla have been previously associated with sleep quality in humans, and there is growing evidence that members of these phyla may modulate circadian rhythm [65] and food intake [66], both of which impact sleep quality. Specifically, Benedict and colleagues found that partial sleep deprivation alters the ratio between these two phyla [38]. Similar findings were reported in mice [37]. However, Zhang and colleagues failed to find any changes in the ratio of these two phyla following sleep restriction [39]. Our study also found that the richness within the Actinobacteria phylum was negatively correlated with the number of awakenings. That is, increased richness within the Actinobacteria contributes to high sleep quality. Similar findings were reported in mice where sleep disruption reduced the percentage of Actinobacteria in the gut microbiome [37]. This contrasts to Benedict and colleagues who found that some members of this phyla increased following sleep restriction in humans [38]. Finally, in contrast to a previous study [40], we found no significant relationships between sleep measures and the richness or diversity within the *Verrucomicrobia* and *Lentisphaera*. This may be owing to differences in sampling methodology (self-report vs. actigraphy). In the previous study [40], the Pittsburgh Sleep Quality Index was used to determine sleep quality whereas herein we used actigraphy. Furthermore, the age groups differed significantly between both studies (64.59 ± 7.54 years in the previous study vs. 22.2 ± 3.11 in this study). Given that sleep quality is significantly lower in older adults relative to younger adults [67] it is possible that changes in gut microbiome phyla, along with other physiological changes, contribute to poor sleep quality with age. This warrants future investigation.

Our redundancy analysis revealed several taxa (genera and families) associated with measures of sleep (Fig 3). Previous work [68] examining the relationship between gut microbiome composition and severity of sleep apnea–hypopnea syndrome found that a decrease in the relative abundance of *Sutterella* and *Brevibacterium* generally coincided with an increase in severity of the disease state. In congruence with these previous findings, our study found that there was a negative correlation between these genera, and the number of awakenings. In addition, the same previous study [68] found an increase in the relative abundance in the *Lachnospiraceae* (family) as the severity of sleep apnea–hypopnea syndrome increased. This also generally agrees with our findings as the *Lachnospiraceae* were, on average, negatively correlated with sleep efficiency and total sleep time. This may indicate that similar genera/families have wide ranging effects on sleep, both in disease and non-disease states.

There is growing interest in identifying the metabolites produced by bacteria that interface through the BGMA. Several human gut associated species in the Bacteroidetes [69], Actinobacteria and Firmicutes [70] phyla produce γ-aminobutyric acid (GABA), a neurotransmitter that promotes sleep [71]. Our results indicate that the diversity and/or richness of these phyla are generally correlated with healthy sleep (e.g., high sleep efficiency, low WASO, low number of awakenings). At the taxa level, *Corynebacterium* have been previously reported to have the metabolic capability to synthesize serotonin, whereas some genera identified positively (*Sutterella*, *Neisseria*) and negatively (e.g., *Blautia*, *Parasutterealla*) correlated with measures of quality sleep do not [5]. This might allude to an important role of the *Corynebacterium* in promoting sleep as serotonin modulates sleep [72], and gut bacteria produce serotonin that appears to interface through the BGMA [73]. Interestingly, serotonin has been previously reported to increase synthesis of IL-6 in some human cell types [74], and that increased IL-6 has been associated with poor cognitive and emotional performance [75]. Moreover, the ability of *Corynebacterium* and *Brevibacterium* to produce the somnogenic factor glutamate has been noted previously [76]. We note that both taxa are negatively correlated with number of awakenings (Fig 3). Finally, our analysis revealed that several taxa from the short chain fatty acid (SCFA) producing Lachnospiraceae family [77], including *Blautia*, *Coprococcus* and *Oribacterium*, are negatively correlated with healthy sleep. While our literature review failed to identify species within this family that produce metabolites that promote wakefulness or reduced sleep quality, a recent study has shown that SCFA produced in the murine gut microbiome peak in concentration at the beginning of the dark period and can otherwise influence circadian rhythm [78]. It remains unclear as to if SCFA produced from the Lachnospiraceae family influences sleep quality, either positively or negatively, in humans. It is important to note that a major caveat of our current research is that we cannot pinpoint directionality of interactions through correlation such as this. Nevertheless, while the aforementioned link is plausible, additional studies are required to elucidate the role that the gut microbiome has in producing and regulating serotonin, and other sleep modulating metabolites, and their direct influence on the immune system and neurobehavioral performance.

It is also important to note that our study was limited to males and, therefore, we cannot be certain of the extent to which our findings apply to women. Previous work from our group showed that sleep loss also increases inflammation in young women [79]. In general, we would expect similar, or even more pronounced, findings in women since the consequences of sleep loss accumulate more quickly in women compared to men [80, 81] and women are at a higher risk than men for sleep loss-related mortality [82]. However, it appears that gender can affect microbiome composition (e.g., [83]), which may result in different associations between taxa and measures of sleep. Nevertheless, we expect that the results are repeatable in men given that the major findings were sufficiently robust as to yield statistical significance at the conventional levels with good effect sizes.

In summary, our results show a novel association between sleep health and gut microbiome diversity. Moreover, we found that IL-6 is as an important player in the sleep-gut microbiome relationship. Finally, we identified several specific phyla and taxa that are related to sleep health, which holds the promise for improved sleep via manipulation of the gut microbiome.

## Supporting information

**S1 Fig. Sleep efficiency is significantly and positively correlated with microbiome diversity.**
a) Pearson correlation analysis of richness and sleep efficiency (ρ = 0.479, P = 0.001). In all panels, dotted line is a linear line plotted through the data.
b) Pearson correlation analysis of Shannon diversity and sleep efficiency (ρ = 0.643, P = 0.001).

c) Pearson correlation analysis of inverse Simpson diversity and sleep efficiency (ρ = 0.540, P = 0.009).
(PNG)

**S2 Fig. Microbiome diversity and measures of sleep are significantly correlated with correct abstract matching.** a) Pearson correlation analysis richness (ρ = 0.489, P = 0.015), Shannon diversity (ρ = 0.607, P = 0.002), and inverse Shannon diversity (ρ = 0.501, P = 0.013) with the number of correct abstract matching responses. In all panels, dotted line is a linear line plotted through the data.
b) Pearson correlation analysis of sleep efficiency (ρ = 0.405, P = 0.044) and WASO (ρ = -0.424, P = 0.035) with correct abstract matching responses.
(PNG)

**S3 Fig. IL-6 correlates significantly with microbiome diversity, measures of sleep and cognitive performance tasks.** a) Pearson correlation analysis of IL-6 with richness (ρ = 0.612, P = 0.001), Shannon diversity (ρ = 0.508, P = 0.011) and inverse Simpsons diversity (ρ = 0.521, P = 0.009. In all panels, dotted line is a linear line plotted through the data.
b) Pearson correlation analysis of IL-6 with time in bed (hrs, ρ = 0.439, P = 0.032) and total sleep time (ρ = 0.476, P = 0.019).
c) Pearson correlation analysis of IL-6 with psychomotor vigilance (ρ = 0.469, P = 0.016), perceived rejection (ρ = 0.451, P = 0.024), working memory (ρ = -0.388, P = 0.045), and meaning and purpose (ρ = -0.507, P = 0.010).
(PNG)

**S4 Fig. Pearson correlation analysis of richness and risk decision making (ρ = -0.461, P = 0.023).**
(PNG)

**S1 Table. Pearson correlation analysis of IL-1β with measures of sleep and microbiome diversity.**
(DOCX)

**S2 Table. Correlation coefficients and P values of associations between measures of sleep, IL-6 and cognition/emotion in our interaction network, and bacterial taxa.**
(DOCX)

## Acknowledgments

This research was supported by a Presidents Faculty Research and Development Grant #335411 through Nova Southeastern University.

## Author Contributions

**Conceptualization:** Robert P. Smith, Cole Easson, Jose V. Lopez, Jaime L. Tartar.

**Data curation:** Robert P. Smith, Cole Easson, Sarah M. Lyle, Ritishka Kapoor, Chase P. Donnelly, Eileen J. Davidson, Esha Parikh, Jaime L. Tartar.

**Formal analysis:** Robert P. Smith, Cole Easson, Chase P. Donnelly, Jose V. Lopez, Jaime L. Tartar.

**Funding acquisition:** Robert P. Smith, Jose V. Lopez, Jaime L. Tartar.

**Investigation:** Sarah M. Lyle, Ritishka Kapoor, Chase P. Donnelly, Eileen J. Davidson, Esha Parikh, Jaime L. Tartar.

**Methodology:** Robert P. Smith, Sarah M. Lyle, Ritishka Kapoor, Esha Parikh, Jose V. Lopez, Jaime L. Tartar.

**Project administration:** Robert P. Smith, Eileen J. Davidson, Jose V. Lopez, Jaime L. Tartar.

**Resources:** Cole Easson, Jose V. Lopez, Jaime L. Tartar.

**Software:** Jose V. Lopez, Jaime L. Tartar.

**Supervision:** Robert P. Smith, Jose V. Lopez, Jaime L. Tartar.

**Writing – original draft:** Robert P. Smith.

**Writing – review & editing:** Cole Easson, Sarah M. Lyle, Ritishka Kapoor, Chase P. Donnelly, Esha Parikh, Jose V. Lopez, Jaime L. Tartar.

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
