## [Decision Letter · Decision Letter 0]

13 Aug 2019

PONE-D-19-19430

Gut microbiome diversity is associated with sleep physiology in humans.

PLOS ONE

Dear Dr. Tartar,

Thank you for submitting your manuscript to PLOS ONE. After careful consideration, we feel that it has merit but does not fully meet PLOS ONE’s publication criteria as it currently stands. Therefore, we invite you to submit a revised version of the manuscript that addresses the points raised during the review process.

Please note that both reviewers have raised a few pertinent questions that are important to address. If you feel you can address the suggestions of the reviewers adequately, then please resubmit by the deadline as mentioned below.

We would appreciate receiving your revised manuscript by Sep 27 2019 11:59PM. To enhance the reproducibility of your results, we recommend that if applicable you deposit your laboratory protocols in protocols.io, where a protocol can be assigned its own identifier (DOI) such that it can be cited independently in the future. For instructions see: http://journals.plos.org/plosone/s/submission-guidelines#loc-laboratory-protocols

We look forward to receiving your revised manuscript.

Kind regards,

Palok Aich, PhD

Academic Editor

PLOS ONE

Journal Requirements:

3. Please provide additional details regarding participant consent. In the ethics statement in the Methods and online submission information, please ensure that you have specified (1) whether consent was informed and (2) what type you obtained (for instance, written or verbal). If your study included minors, state whether you obtained consent from parents or guardians. If the need for consent was waived by the ethics committee, please include this information.

We also ask that you please include in your methods section, the date range of patient recruitment and data collection.

4. Our internal editors have looked over your manuscript and determined that it is within the scope of our The Microbiome Across Biological Systems Call for Papers. This collection of papers is headed by a team of Guest Editors for PLOS ONE: Zaid Abdo, Colorado State University, USA; Sanjay Chotrimall, Lee Kong Chian School of Medicine, Nanyang Technological University, Singapore; Noelle Noyes, University of Minnesotta, USA;  Pankaj Trivedi, Colorado State University, USA; and Thomas Dawson, A*STAR, Singapore. The Collection will encompass a diverse range of research articles about microbiomes and human health, the natural and built environment, and new technologies used to study microbiomes. Additional information can be found on our announcement page: https://collections.plos.org/s/microbiome. If you would like your manuscript to be considered for this collection, please let us know in your cover letter and we will ensure that your paper is treated as if you were responding to this call. If you would prefer to remove your manuscript from collection consideration, please specify this in the cover letter.

Reviewers' comments:

Reviewer's Responses to Questions

**Comments to the Author**

1. Is the manuscript technically sound, and do the data support the conclusions?

Reviewer #1: Yes

Reviewer #2: Yes

2. Has the statistical analysis been performed appropriately and rigorously? 

Reviewer #1: Yes

Reviewer #2: Yes

3. Have the authors made all data underlying the findings in their manuscript fully available?

Reviewer #1: Yes

Reviewer #2: Yes

4. Is the manuscript presented in an intelligible fashion and written in standard English?

Reviewer #1: Yes

Reviewer #2: Yes

5. Review Comments to the Author

Reviewer #1: The authors evaluate the association between gut microbiome diversity and sleep physiology. The study group consisted of male subjects and evaluated the participants sleep using and Actiwatch, cognition emotion, IL-1, IL-6 and cortisol by saliva samples and the microbiome using next generation sequencing of fecal samples.

The main results from the experiment were that microbiome diversity is positively correlated with sleep efficiency. In addition IL-6 is correlated to micobiome diversity and some measurements of sleep. Bacteroidetes and Firmicutes were associated with sleep efficiency IL-6 and abstract thought.

Questions to the authors.

Could the authors comment on the role of gender and if the results would be similar in females?

Could the authors comment on the reproducibility of the data if the trial was repeated?

As this study looks at correlation it does not mean causality. The authors preliminary study potentially opens up avenues to further evaluate the role of the microbiome in sleep efficiency. This is important work as issues with sleep in individuals is a health problem for a great percentage of the population. Given the specific bacteria identified which may play a role in sleep efficiency, can the authors comment on fecal transplant and if this could be a potential treatment to improve sleep efficiency.

Reviewer #2: In this manuscript, Smith et.al. made an intelligent effort to identify the correlations of gut microbiome diversity with sleep deprivation in normal individuals. They have also included a spotlight on inflammatory marker, interleukin-6 (IL-6) to recognize the association of of gut microbiome diversity and inflammation. Overall, the manuscript provides interesting data showing correlation of different microbiome taxa with markers of cognition and sleep apnea. There are few minor concerns:

Methods and Results

1) The inflammatory markers were assessed in saliva. The sleep deprivation has also been shown to modulate the inflammatory markers in peripheral blood. The authors should consider measuring the level of inflammatory markers including IL-6 in plasma/serum of the participants. The results would provide more stringent assessment of correlation between inflammation and gut microbiota diversity.

2) It is not clear how the data from Elisa assay for inflammatory markers in saliva were normalized. The drool method of collection generally yields different volume of saliva. It should be normalized to the total protein used for the assay.

Discussion

Some discussions on the potential physiological/biochemical reasoning as to why different microbiota had different relationship with sleep and/or cognitive parameters. Is there any specific cell signaling that could have influence the quality of sleep? A discussion on this matter would be helpful for readers.

6. PLOS authors have the option to publish the peer review history of their article (what does this mean?). If published, this will include your full peer review and any attached files.

Reviewer #1: No

Reviewer #2: No

---

## [Author Response · Author response to Decision Letter 0]

19 Aug 2019

Palok Aich, PhD

Academic Editor

PLOS ONE

August 19, 2019

Dear Dr. Palok Aich, Editorial Board: 

We are re-submitting our manuscript entitled “Gut microbiome diversity is associated with sleep physiology in humans” to be considered for publication in the journal PloS One. 

We thank you and the Reviewers for evaluating our original manuscript and for providing very insightful comments and suggestions. In short, the Reviewers generally appreciated our findings, the quality of the data and the conclusions that we reached. Reviewer 1 has suggested some additional aspects to add to the discussion, while Reviewer 2 has suggested that we provide additional details in the methods section and has suggested we further our discussion of microbial physiology as it relates to sleep. We have addressed all of the comments made by the Reviewers and have incorporated all of their suggestions. 

Please note that we are interested in being considered for “The Microbiome Across Biological Systems Call for Papers.”

With these revisions, in addition to more specific revisions detailed below, we hope that you and the Reviewers will find the manuscript sufficiently improved and suitable for publication. 

Please find a point-by-point response to the Reviewer’s comments below. In the attached cover letter response to each comment below is indicated by the use of bold case, and quotes from the revised proposal are indicated by quotation marks and indentation (below no formatting has been applied). Edits to the original manuscript are red-lined in both the responses and the manuscript.

I look forward to hearing your editorial comments and decision.

Sincerely,

Jaime Tartar and co-authors

Jaime L. Tartar, Ph.D.

Professor of Behavioral Neuroscience

Department of Psychology and Neuroscience 

Nova Southeastern University 

Editorial Comments

We have double-checked out document against the style requirements and make changes to the supplemental figure titles. 

We now put the figure legends for the supporting information at the end of the manuscript. 

3. Please provide additional details regarding participant consent. In the ethics statement in the Methods and online submission information, please ensure that you have specified (1) whether consent was informed and (2) what type you obtained (for instance, written or verbal). If your study included minors, state whether you obtained consent from parents or guardians. If the need for consent was waived by the ethics committee, please include this information.

We now clarify this information as follows:

Recruitment and testing procedures were approved by the Nova Southeastern University (NSU) Institutional Review Board (IRB). All participants received a verbal explanation of the study procedures and signed an NSU IRB-approved written Informed Consent Form.

We also ask that you please include in your methods section, the date range of patient recruitment and data collection.

We now state that:

“Participant recruitment and testing occurred between May of 2017 and March of 2018.” 

4. Our internal editors have looked over your manuscript and determined that it is within the scope of our The Microbiome Across Biological Systems Call for Papers. This collection of papers is headed by a team of Guest Editors for PLOS ONE: Zaid Abdo, Colorado State University, USA; Sanjay Chotrimall, Lee Kong Chian School of Medicine, Nanyang Technological University, Singapore; Noelle Noyes, University of Minnesotta, USA; Pankaj Trivedi, Colorado State University, USA; and Thomas Dawson, A*STAR, Singapore. The Collection will encompass a diverse range of research articles about microbiomes and human health, the natural and built environment, and new technologies used to study microbiomes. Additional information can be found on our announcement page: https://collections.plos.org/s/microbiome. If you would like your manuscript to be considered for this collection, please let us know in your cover letter and we will ensure that your paper is treated as if you were responding to this call. If you would prefer to remove your manuscript from collection consideration, please specify this in the cover letter.

Thank you for letting us know and for your prompt response to our inquiry. We now indicate in the cover letter that we would like our manuscript to be considered for this collection

Reviewer #1: The authors evaluate the association between gut microbiome diversity and sleep physiology. The study group consisted of male subjects and evaluated the participants sleep using and Actiwatch, cognition emotion, IL-1, IL-6 and cortisol by saliva samples and the microbiome using next generation sequencing of fecal samples. The main results from the experiment were that microbiome diversity is positively correlated with sleep efficiency. In addition IL-6 is correlated to microbiome diversity and some measurements of sleep. Bacteroidetes and Firmicutes were associated with sleep efficiency IL-6 and abstract thought.

Questions to the authors.

1. Could the authors comment on the role of gender and if the results would be similar in females?

We appreciate this point and agree that there could be possible sex effects in the outcome measures. We now specifically address this point in the discussion as follows: 

“It is also important to note that our study was limited to males and, therefore, we cannot be certain of the extent to which our findings apply to women. Previous work from our group showed that sleep loss also increases inflammation in young women [79]. In general, we would expect similar, or even more pronounced, findings in women since the consequences of sleep loss accumulate more quickly in women compared to men [80, 81] and women are at a higher risk than men for sleep loss-related mortality [82]. However, it appears that gender can affect microbiome composition (e.g., [83]), which may result in different associations between taxa and measures of sleep.”

2. Could the authors comment on the reproducibility of the data if the trial was repeated?

We now addresses this concern as follows:

“Nevertheless, we expect that the results are repeatable in men given that the major ﬁndings were sufﬁciently robust as to yield statistical signiﬁcance at the conventional levels with good effect sizes.”

3. As this study looks at correlation it does not mean causality. The authors preliminary study potentially opens up avenues to further evaluate the role of the microbiome in sleep efficiency. This is important work as issues with sleep in individuals is a health problem for a great percentage of the population. Given the specific bacteria identified which may play a role in sleep efficiency, can the authors comment on fecal transplant and if this could be a potential treatment to improve sleep efficiency. 

We agree with the Reviewer that we should have discussed the promise of fecal transplants as a mechanism to improve sleep efficiency. Accordingly, we have added the following to the discussion section. 

“For example, previous studies have demonstrated that fecal microbial transplants can improve disorders that are directly linked to the gastrointestinal tract (e.g., recurrent intestinal infections, ulcerative colitis [54, 55]). However, more recent work has demonstrated that such transplant strategies can alter aspects of human physiology that are not directly linked to the gastrointestinal tract but are instead conceivably linked via the BGMA. For example, fecal microbial transplants have been shown to improve cognition in patients suffering from cirrhosis [56], have altered behavior in individuals with autism [57], and have attenuated epileptic seizures [58]. While disruptions to sleep and sleep/wake functions are not considered to be gastrointestinal diseases, these recent studies indicate that fecal microbial transplants may represent a strategy to improve sleep efficiency via the BGMA.” 

Reviewer #2: In this manuscript, Smith et.al. made an intelligent effort to identify the correlations of gut microbiome diversity with sleep deprivation in normal individuals. They have also included a spotlight on inflammatory marker, interleukin-6 (IL-6) to recognize the association of gut microbiome diversity and inflammation. Overall, the manuscript provides interesting data showing correlation of different microbiome taxa with markers of cognition and sleep apnea. There are few minor concerns:

Methods and Results

1) The inflammatory markers were assessed in saliva. The sleep deprivation has also been shown to modulate the inflammatory markers in peripheral blood. The authors should consider measuring the level of inflammatory markers including IL-6 in plasma/serum of the participants. The results would provide more stringent assessment of correlation between inflammation and gut microbiota diversity.

The ELISA kits that we used were specifically developed for analyses in saliva only since we targeted saliva for our analyses during the study planning stage. We find that salivary analyses prevent the non-specific stress effects that can occur during venipuncture (especially in young adult populations who might not have ever experienced a blood draw). In general, saliva samples of cortisol, IL-1β, and IL-6 are reliably determined from saliva. For example, the saliva-serum correlation of cortisol is .91, according to the manufacturer. In addition, since we tested a young, healthy population, saliva was the optimal choice of specimen since plasma levels of pro-inflammatory cytokines in healthy individuals are very low and often below detectable limits. The levels are generally higher, and detectable, in saliva compared to plasma or serum (1-3). Finally there is a positive correlation between plasma and saliva concentrations in the change in IL-1β and IL-6 following stress and, importantly, the variation of the concentration is also positively correlated to the variation in salivary concentration (4). We have now included the sensitivity and range for reach biomarker:

“Final concentrations for the biomarkers were generated by interpolation from the standard curve in µg/dL for cortisol (sensitivity = <0.007, range 0.012-3.000 ug/dL) and pg/mL for IL-1� (sensitivity = <0.37 pg/mL, range 3.13-200 pg/mL) and IL-6 (sensitivity = 0.07 pg/mL, 0 – 100 pg/mL).” 

1. Dinarello, C.A. (1996). Biologic basis for interleukin-1 in disease. Blood, 87(6), 2095-147.

2. Brailo, V., Vucicevic-Boras, V., Lukac, J., Biocina-Lukenda, D., Zilic-Alajbeg, I., Milenovic, A., & Balija, M. (2012). Salivary and serum interleukin 1 beta, interleukin 6 and tumor necrosis factor alpha in patients with leukoplakia and oral cancer. Med Oral Patol Oral Cir Bucal, 17(1), e10-5.

3. Wong, H.L., Pfeiffer, R.M., Fears. T.R., Vermeulen, R., Ji, S., & Rabkin, C.S. (2008). Reproducibility and correlations of multiplex cytokine levels in asymptomatic persons. Canc Epidemiol Biomarkers Prev, 17(12), 3450-56.

4. Slavish, D. C., Graham-Engeland, J. E., Smyth, J. M., & Engeland, C. G. (2015). Salivary markers of inflammation in response to acute stress. Brain, behavior, and immunity, 44, 253-269.

2) It is not clear how the data from Elisa assay for inflammatory markers in saliva were normalized. The drool method of collection generally yields different volume of saliva. It should be normalized to the total protein used for the assay.

We normalized this data by collecting the same amount of saliva from each participant as participants were instructed to fill the collection tubes up to pre-marked 1 mL line. Samples were analyzed according to the manufacturer’s instructions, which normalizes the quantity of saliva used in the assay. Importantly, cortisol, IL-1β, and IL-6 are not sensitive to salivary flow rates. We also measured the samples in duplicate. 

The largest source of error is introduced at the time of collection due to food particles than can alter the pH. Accordingly, all participants were asked to refrain from eating 1 hour prior to sample collection and rinsed their mouths with water ~10 minutes before sample collection. 

We note that standardizing immunomarkers by other means (e.g., total protein) is not common practice when using this assay. An abundance of previously published literature across disciplines supports this notion. Examples include, but are not limited to, the following: 

Menon, M. M., Balagopal, R. V., Sajitha, K., Parvathy, K., Sangeetha, G. B., Arun, X. M., & Sureshkumar, J. (2016). Evaluation of salivary interleukin-6 in children with early childhood caries after treatment. Contemporary clinical dentistry, 7(2), 198–202. doi:10.4103/0976-237X.183059

Gill, J., Vythilingam, M., & Page, G. G. (2008). Low cortisol, high DHEA, and high levels of stimulated TNF-alpha, and IL-6 in women with PTSD. Journal of traumatic stress, 21(6), 530–539. doi:10.1002/jts.20372

O'Donovan, A., Hughes, B. M., Slavich, G. M., Lynch, L., Cronin, M. T., O'Farrelly, C., & Malone, K. M. (2010). Clinical anxiety, cortisol and interleukin-6: evidence for specificity in emotion-biology relationships. Brain, behavior, and immunity, 24(7), 1074–1077. doi:10.1016/j.bbi.2010.03.003

Ives, S. J., Blegen, M., Coughlin, M. A., Redmond, J., Matthews, T., & Paolone, V. (2011). Salivary estradiol, interleukin-6 production, and the relationship to substrate metabolism during exercise in females. European journal of applied physiology, 111(8), 1649-1658.

Hamilton, L. D., Rellini, A. H., & Meston, C. M. (2008). Cortisol, sexual arousal, and affect in response to sexual stimuli. The journal of sexual medicine, 5(9), 2111–2118. doi:10.1111/j.1743-6109.2008.00922.x

Rellini, A. H., Hamilton, L. D., Delville, Y., & Meston, C. M. (2009). The cortisol response during physiological sexual arousal in adult women with a history of childhood sexual abuse. Journal of traumatic stress, 22(6), 557–565. doi:10.1002/jts.20458

Oberlander, T. F., Weinberg, J., Papsdorf, M., Grunau, R., Misri, S., & Devlin, A. M. (2008). Prenatal exposure to maternal depression, neonatal methylation of human glucocorticoid receptor gene (NR3C1) and infant cortisol stress responses. Epigenetics, 3(2), 97-106.

Sturge-Apple, M. L., Davies, P. T., Cicchetti, D., & Manning, L. G. (2012). Interparental violence, maternal emotional unavailability and children's cortisol functioning in family contexts. Developmental psychology, 48(1), 237.

D'Anna-Hernandez, K. L., Ross, R. G., Natvig, C. L., & Laudenslager, M. L. (2011). Hair cortisol levels as a retrospective marker of hypothalamic-pituitary axis activity throughout pregnancy: comparison to salivary cortisol. Physiology & behavior, 104(2), 348–353. doi:10.1016/j.physbeh.2011.02.041

Taken together, while we did not standardize to total protein, we standardized the amount of saliva collected and the amount of saliva used. As such, our measurements of immunomarkers from saliva are technically sound. 

Discussion

Some discussions on the potential physiological/biochemical reasoning as to why different microbiota had different relationship with sleep and/or cognitive parameters. Is there any specific cell signaling that could have influence the quality of sleep? A discussion on this matter would be helpful for readers.

We thank the reviewer for this comment. In the last paragraph of our original submission, we discussed the ability of Corynebacterium to synthesize serotonin, which may interface through the BGMA. To address the Reviewer’s comment, we expanded this paragraph to include bacteria that synthesize the somnogenic factors GABA and glutamate, as well as the ability of the Lachnospiraceae to produce short chain fatty acids, which may influence circadian rhythm in humans. The final paragraph of our manuscript now reads as follows:

“There is growing interest in identifying the metabolites produced by bacteria that interface through the BGMA. Several human gut associated species in the Bacteroidetes [69], Actinobacteria and Firmicutes [70] phyla produce γ-aminobutyric acid (GABA), a neurotransmitter that promotes sleep [71]. Our results indicate that the diversity and/or richness of these phyla are generally correlated with healthy sleep (e.g., high sleep efficiency, low WASO, low number of awakenings). At the taxa level, Corynebacterium have been previously reported to have the metabolic capability to synthesize serotonin, whereas some genera identified positively (Sutterella, Neisseria) and negatively (e.g., Blautia, Parasutterealla) correlated with measures of quality sleep do not [5]. This might allude to an important role of the Corynebacterium in promoting sleep as serotonin modulates sleep [72], and gut bacteria produce serotonin that appears to interface through the BGMA [73]. Interestingly, serotonin has been previously reported to increase synthesis of IL-6 in some human cell types [74], and that increased IL-6 has been associated with poor cognitive and emotional performance [75]. Moreover, the ability of Corynebacterium and Brevibacterium to produce the somnogenic factor glutamate has been noted previously [76]. We note that both taxa are negatively correlated with number of awakenings (Table 2). Finally, our analysis revealed that several taxa from the short chain fatty acid (SCFA) producing Lachnospiraceae family [77], including Blautia, Coprococcus and Oribacterium, are negatively correlated with healthy sleep. While our literature review failed to identify species within this family that produce metabolites that promote wakefulness or reduced sleep quality, a recent study has shown that SCFA produced in the murine gut microbiome peak in concentration at the beginning of the dark period and can otherwise influence circadian rhythm [78]. It remains unclear as to if SCFA produced from the Lachnospiraceae family influences sleep quality, either positively or negatively, in humans. It is important to note that a major caveat of our current research is that we cannot pinpoint directionality of interactions through correlation such as this. Nevertheless, while the aforementioned link is plausible, additional studies are required to elucidate the role that the gut microbiome has in producing and regulating serotonin, and other sleep modulating metabolites, and their direct influence on the immune system and neurobehavioral performance.”

---

## [Decision Letter · Decision Letter 1]

29 Aug 2019

[EXSCINDED]

Gut microbiome diversity is associated with sleep physiology in humans

PONE-D-19-19430R1

Dear Dr. Tartar,

We are pleased to inform you that your manuscript has been judged scientifically suitable for publication and will be formally accepted for publication once it complies with all outstanding technical requirements.

With kind regards,

Palok Aich, PhD

Academic Editor

PLOS ONE

Additional Editor Comments (optional):

Reviewers' comments:

Reviewer's Responses to Questions

**Comments to the Author**

1. If the authors have adequately addressed your comments raised in a previous round of review and you feel that this manuscript is now acceptable for publication, you may indicate that here to bypass the “Comments to the Author” section, enter your conflict of interest statement in the “Confidential to Editor” section, and submit your "Accept" recommendation.

Reviewer #1: All comments have been addressed

Reviewer #2: All comments have been addressed

2. Is the manuscript technically sound, and do the data support the conclusions?

Reviewer #1: Yes

Reviewer #2: Yes

3. Has the statistical analysis been performed appropriately and rigorously? 

Reviewer #1: Yes

Reviewer #2: Yes

4. Have the authors made all data underlying the findings in their manuscript fully available?

Reviewer #1: Yes

Reviewer #2: Yes

5. Is the manuscript presented in an intelligible fashion and written in standard English?

Reviewer #1: Yes

Reviewer #2: Yes

6. Review Comments to the Author

Reviewer #1: The authors have sufficiently responded to the reviewers comments and the paper is suitable for publication.

Reviewer #2: The authors have adequately answered the concerns raised during the first review. The normalization of the inflammatory markers with respect to equal volume of collected saliva is acceptable. The authors have also expanded the discussion on the differential sleep physiology based to various microbiota.

7. PLOS authors have the option to publish the peer review history of their article (what does this mean?). If published, this will include your full peer review and any attached files.

Reviewer #1: No

Reviewer #2: No

---

## [Editor Report · Acceptance letter]

11 Sep 2019

PONE-D-19-19430R1 

Gut microbiome diversity is associated with sleep physiology in humans 

Dear Dr. Tartar:

I am pleased to inform you that your manuscript has been deemed suitable for publication in PLOS ONE. Congratulations! Your manuscript is now with our production department. 

With kind regards,

on behalf of

Dr. Palok Aich 

Academic Editor

PLOS ONE